# MicroRNAs Derived from Extracellular Vesicles: Keys to Understanding SARS-CoV-2 Vaccination Response in Cancer Patients?

**DOI:** 10.3390/cancers15164017

**Published:** 2023-08-08

**Authors:** Beatriz Almeida, Tânia R. Dias, Ana Luísa Teixeira, Francisca Dias, Rui Medeiros

**Affiliations:** 1Molecular Oncology and Viral Pathology Group, Research Center of IPO Porto (CI-IPOP) & RISE@CI-IPOP (Health Research Network), Portuguese Oncology Institute of Porto (IPO Porto), Porto Comprehensive Cancer Center (Porto.CCC), 4200-072 Porto, Portugal; beatriz.almeida@ipoporto.min-saude.pt (B.A.); tania.dias@ipoporto.min-saude.pt (T.R.D.); ana.luisa.teixeira@ipoporto.min-saude.pt (A.L.T.); ruimedei@ipoporto.min-saude.pt (R.M.); 2Department of Chemistry, University of Aveiro, Campus Universitário de Santiago, 3810-193 Aveiro, Portugal; 3Abel Salazar Institute for the Biomedical Sciences (ICBAS), University of Porto, 4050-513 Porto, Portugal; 4Laboratory Medicine, Clinical Pathology Department, Portuguese Oncology Institute of Porto (IPO-Porto), Porto Comprehensive Cancer Center (Porto.CCC), 4200-072 Porto, Portugal; 5Biomedicine Research Center (CEBIMED), Research Inovation and Development Institute (FP-I3ID), Faculty of Health Sciences, Fernando Pessoa University (UFP), 4249-004 Porto, Portugal; 6Research Department, Portuguese League against Cancer Northern Branch (LPCC-NRN), 4200-172 Porto, Portugal

**Keywords:** extracellular vesicles, microRNAs, SARS-CoV-2

## Abstract

**Simple Summary:**

Despite the declaration of the end of the COVID-19 pandemic in May 2023, several important questions regarding its impact on public health remain unanswered. Specifically, the clinical trials for COVID-19 vaccines did not include cancer patients, leaving a knowledge gap about how vaccination affects this vulnerable population. Therefore, it is crucial to identify specific biomarkers that can help us understand and stratify cancer patients according to their SARS-CoV-2 vaccination responsiveness. Thus, a literature review was conducted to investigate the known extracellular vesicle-derived microRNAs involved in SARS-CoV-2 infection and how many that were already deregulated in the cancer context could also be involved in the vaccination response. However, further in-depth studies are necessary to fully comprehend and validate the role of these EV-miRNAs as potential biomarkers of cancer patients’ response to vaccination.

**Abstract:**

Severe acute respiratory syndrome coronavirus 2 (SARS-CoV-2) provoked a global pandemic identified as coronavirus disease (COVID-19), with millions of deaths worldwide. However, several important questions regarding its impact on public health remain unanswered, such as the impact of vaccination on vulnerable subpopulations such as cancer patients. Cytokine storm and a sustained inflammatory state are commonly associated with immune cell depletion, being manifested in most immunocompromised individuals. This strong immunosuppression can lead to a dysfunctional antiviral response to natural viral infection and compromised vaccination response. Extracellular vesicles (EVs) are membrane-bound vesicles released from cells that are involved in intercellular communication. EVs carry various molecules including microRNAs that play a crucial role in COVID-19 pathophysiology, influencing cellular responses. This review summarizes the state of the art concerning the role of EV-derived miRNAs in COVID-19 infection and their potential use as prognosis biomarkers for vaccination response in cancer patients.

## 1. Introduction

### 1.1. The Problematic behind COVID-19

Coronavirus disease 2019 (COVID-19), caused by novel severe acute respiratory syndrome coronavirus 2 (SARS-CoV-2) infection, was identified in December of 2019 in Wuhan, China. This viral infection caused a global pandemic with more than 767 million confirmed cases and more than 6 million deaths worldwide [1]. SARS-CoV-2 is classified as a *Nidovirales*, belonging to the subfamily *Orthocoronavirinae*, with four genera, namely, *Alphacoronavirus* (α), *Betacoronavirus* (β), *Gammacoronavirus* (γ), and *Deltacoronavirus* (δ) [2]. SARS-CoV-2 appears to have similarity with the other members of its subfamily, since its genome has 79% sequence homology with SARS-CoV and 50% homology with MERS-CoV [3]. In general, the genome of coronaviruses (CoVs) consists of a single-stranded positive-sense RNA molecule, approximately 30 kb in length, which is the largest genome among all known RNA viruses. The genome of this virus is characterized as being encapsidated by a nucleoprotein (N), forming a complex recognized as a nucleocapsid [4,5]. According to Ke et al., *Betacoronaviruses* exhibit a spherical structure and are classified as enveloped RNA viruses with a diameter ranging from 80 to 160 nm [5,6,7]. The genome of this virus can originate 16 nonstructural proteins (nsp) and 4 structural proteins: the spike glycoprotein (S), envelope protein (E), membrane protein (M), and nucleocapsid phosphoprotein (N). On the surface of the virus, spike protein trimers are present. These spike proteins, also known as S proteins, play a vital role in the infection process [8], since they are crucial for receptor recognition, viral attachment, and entry into host cells in all human coronaviruses (HCoVs). Given their essential role, they are a primary focus of COVID-19 vaccine and therapeutic research [9]. The mechanism of infection initiates with the binding of S trimers of SARS-CoV-2 to the host cell receptor, angiotensin-converting enzyme 2 (ACE2), a crucial receptor of viral infection [10,11,12]. After the connection of these two molecular components on the surface of host cells, a subsequent viral uptake and fusion takes place. The spike protein passes on to a dramatic structural rearrangement from the prefusion form to the postfusion form, provoked essentially by transmembrane serine protease 2 (TMPRSS2) and furin of the host [5,13]. More specifically, the S protein undergoes proteolytic cleavage at the S1/S2 site and subsequently at the S2’ site, preparing it for membrane fusion. Once inside the host cell and during viral infection, coronaviruses’ machinery extensively remodels the internal membrane organization of the cell, generating viral replication organelles in order to spread the virus into other cells. The S protein and the other membrane proteins of the virus, such as M and E proteins, are inserted into membranes of the endoplasmic reticulum (ER) and are consequently trafficked to the ER golgi intermediate compartment (ERGIC) [5,14,15]. The mature virions are then released from the host cell, allowing for further infection and the spread of the disease.

The epithelial cells of the respiratory tract have abundant ACE2 receptors, promoting the beginning of viral infection in this particular site. In most of the SARS-CoV-2-infected carriers, the virus is contained in the upper respiratory tract, resulting in either no symptoms or mild symptoms. The main clinical features of COVID-19 are fever, cough, dyspnoea, anosmia, and dysgeusia [16]. However, some individuals have COVID-19 severe respiratory and nonrespiratory manifestations, including multiorgan failure and shock among patients with severe and fatal disease [17]. In these severe cases, there are reports that describe SARS-CoV-2 infection as a cause of pneumonia and excessive inflammation, which can lead to acute respiratory distress syndrome (ARDS). ARDS may then lead to organ damage, a hyperinflammatory state, and lymphocytopenia, a hallmark that appears as both a signature and prognosis of disease severity [18,19]. Cytokine storm and a sustained inflammatory state are events commonly associated with immune cell depletion, being manifested in most immunocompromised individuals [20,21]. This strong immunosuppression can lead to a dysfunctional antiviral response to both natural viral infection and compromised vaccination response.

To combat the widespread impact of COVID-19, the development of vaccines became an urgent priority. Currently, vaccination and antiviral drugs remain the primary focus for biotech and pharmaceutical companies in their efforts to fight this pandemic [22]. It is essential to identify specific subpopulations that may experience greater benefits from one approach over the other or may require a combination of both approaches to effectively control the spread of the virus. This knowledge will guide us in implementing targeted and personalized strategies to combat the pandemic more efficiently.

### 1.2. Vaccines Development against COVID-19

Community-level immunity, acquired through infection or vaccination, is crucial to control the pandemic as the virus continues to circulate. Since vaccination is currently the most effective strategy for the mitigation of the COVID-19 pandemic, several types of vaccines were developed [23]. They are based on viral vectors, nucleic acids, protein subunits, inactivated or attenuated viruses, peptides, or viruslike particles (VLP) [24,25].

The main vaccines available worldwide are BNT162b, produced by Pfizer (Manhattan, NY, USA), and mRNA-1273, produced by Moderna (New England, MA, USA), both of which are based on lipid nanoparticle delivery of mRNA encoding a prefusion-stabilized form of spike protein derived from SARS-CoV-2 [26,27]; adenovirus-based vaccines AZD1222, manufactured by AstraZeneca (Cambridge, UK), and Ad26.COV2.S, produced by Johnson & Johnson (New Brunswick, Nova Jersey, EUA); a nanoparticle-based vaccine NVX-CoV2373, manufactured by Novavax (Gaithersburg, MD, USA); and an inactivated protein vaccine manufactured by CoronaVac and Sinopharm (Pequim, China) [28]. These last four vaccines demonstrated reduced overall efficacy, while the first two were the main vaccines administered, with more than 94% efficacy in preventing COVID-19 [29,30,31,32,33].

The mRNA vaccines are delivered in various formats, including encapsulation by delivery carriers such as lipid nanoparticles, polymers, and peptides, free mRNA in solution, and ex vivo through dendritic cells [34,35,36]. Particularly, the effectiveness of mRNA vaccines is enhanced when they are delivered using lipid nanoparticles, since these nanoparticles have the ability to specifically target dendritic cells and, consequently, stimulate immune responses. As a result, the delivery of mRNA vaccines using lipid nanoparticles stimulates both cellular and humoral immune responses, promoting a better vaccine efficacy [37,38]. Nonetheless, lipid nanoparticles used for vaccine delivery possess similarities to endogenous lipid nanoparticles found in human organisms. This feature makes them favourable carriers for mRNA vaccination since the immune system recognizes the lipid nanoparticle with the mRNA of the viral target without identifying it as a foreign entity that needs to be eliminated. mRNA vaccines trigger the immune system to produce neutralizing antibodies (NAbs) against SARS-CoV-2 spike proteins [39,40,41]. Manipulating mRNA to enable host cells to produce viral antigenic protein fragments offers several advantages. One key benefit is that producing mRNA is easier compared to the traditional method of whole viral inactivation. In addition to that, transcription reaction is easy to conduct, has a high yield, and can be scaled up [42,43,44]. Moreover, mRNA vaccines enable the synthesis of antigen proteins in situ, eliminating the need for protein purification and long-term stabilization, which are challenging for some antigen transportation; furthermore, storage of mRNA may be easier than protein-based vaccines since RNA, if protected properly against ribonucleases (RNases), is less prone to degradation compared to proteins [40,43,45].

The establishment of immunity against SARS-CoV-2 has become a central focus of current research efforts worldwide [36,46,47]. Natural immunity following infection and vaccine-generated immunity provide two different pathways to immunity against the disease [48]. mRNA vaccines have demonstrated significant protection against severe COVID-19 disease. In fact, findings from human trials of Pfizer/BioNTech and Moderna vaccines suggest 95% maximal protection within 1 to 2 months after the second vaccine dose, including against several circulating variants of concern [49,50]. The presence of neutralizing antibodies is currently used as a surrogate indicator of immunity. The mRNA-1273 and BNT162b2 vaccines trigger the immune system to produce neutralizing antibodies (NAbs) against SARS-CoV-2 spike proteins, proposed as the major correlate of protection, promoting a robust germinal center response in humans, resulting in memory B cells that are specific for both the full-length SARS-CoV-2 spike protein and the spike receptor-binding domain (RBD) [51]. Both mRNA vaccines induce potent and durable neutralizing antibodies as early as 10 days and last up to 8 months after the first dose of vaccination [49,52].

Delivering a conformationally accurate part of the virus is crucial to induce antibody-mediated immunity in vaccines. Many factors influence the immune response elicited by vaccines. The most obvious and major factor is the vaccine itself. This includes the vaccine type, the pathogen used in the formulation, the type of adjuvant, and the dose of the vaccine used for immunization. The vaccination schedule as well as the route and site of vaccine delivery also play a crucial role [53]. There are also several host factors that have an impact on the immunogenicity of the vaccine and need to be considered when defining vaccination calendars and booster doses. These include age, gender, genetic aspects, as well as the presence of comorbidities, most notably, diabetes [54], hypertension [55], cardiovascular diseases [56], and cancer [57,58].

### 1.3. Cancer Patients and mRNA Vaccines against COVID-19

Cancer has revealed itself as an emergent public health problem. Nowadays, it is the first and the third cause of death in the developed countries and in the less developed countries, respectively [59].

In the case of SARS-CoV-2 infection, cancer patients are at higher risk of severe manifestations and thus have been considered as a high-priority group for COVID-19 vaccination, although the evidence regarding the immunogenicity and safety of COVID-19 vaccines for this frail population is very limited [60]. The clinical trials for vaccine approval largely excluded immunocompromised individuals, including patients receiving immunosuppressive therapies to control chronic inflammatory conditions, patients with primary immunodeficiencies, organ transplant recipients, and cancer patients receiving cytotoxic chemotherapy [61]. In fact, in the phase III trial of the BNT162b2 vaccine, cancer patients were mostly excluded, and only small cohort studies enrolling cancer patients have been available to date [46]. Concern has been particularly high about the impact of COVID-19 vaccination on cancer patients, since a study from the COVID-19 Cancer Consortium showed a 13% 30-day all-cause mortality from COVID-19, which is 10 to 30 times greater than that observed in the general population. Importantly, the investigators noted a higher risk of death in patients with active cancer [46]. Moreover, immune responses after receipt of SARS-CoV-2 messenger RNA (mRNA) vaccines have been found to be decreased in immunocompromised patients because of the use of immunosuppressive therapeutic approaches [62].

### 1.4. Necessity of Biomarkers of Immune Response to COVID-19 Vaccines: Can EV-miRNAs Help in Cancer Patients’ Immune Response Stratification?

Since there is little information on the development of the immune response in cancer patients after SARS-CoV-2 vaccination, the identification of biomarkers that can stratify cancer patients into responders and nonresponders to mRNA vaccination seems essential for more precise treatment in this subpopulation. Therefore, there are reports that refer to the encapsulated miRNAs from extracellular vesicles (EVs) as potential biomarkers of many clinical statuses, particularly in cancer prognosis, since these EV-miRNAs have an important role as intercellular messengers [63,64,65]. Most cell types in the body typically release around 6000 to 15,000 EVs per day [66]. Upon secretion to the extracellular space, EVs may be taken up by neighbouring cells or diffuse into the systemic circulation, potentially reaching tissues distant from their cells of origin [67,68]. Through the incorporation of the EV cargo, the phenotype of the recipient cell may be altered, as can be demonstrated through the transfer of malignant properties from one cell to another during cancer progression [68,69,70]. Regarding SARS-CoV-2 viral infection, there are some studies revealing an important function of extracellular vesicles [10,71]; specifically, EVs that contain an ACE2 receptor serve as a defence mechanism against SARS-CoV-2 infection [72].

Due to their importance, EVs became one of the main fields of investigation in vaccine development, since they are described as nanoparticles with a lipid bilayer membrane that can carry various components, including proteins, lipids, and nucleic acids [73].

EVs also contain significant amounts of a specific nucleic acid, microRNA (miRNA). These molecules are characterized as 22-nucleotide noncoding small RNAs, which play a pivotal role in gene regulation by silencing post-transcriptional gene expression through mRNA targeting [74]. This interaction of miRNAs with target mRNAs can lead to the suppression or degradation of these mRNAs, preventing the translation of specific proteins. By regulating a large repertoire of both proto-oncogenes and tumor-suppressor genes, miRNAs play an important role in oncogenesis, and it is also well established that miRNA profiles are highly deregulated in cancer. In the SARS-CoV-2 infection context, there are studies evidencing the crucial role of microRNAs in distinguish the different symptoms and complications of SARS-CoV-2 through the alteration of their expression in COVID-19 patients [75]. For example, Garg et al. [76] investigated the differential expression of microRNAs in influenza–ARDS and COVID-19 disease, reporting that upregulation of miR-21, miR-155, miR-208a, and miR-499 was more markedly related to inflammation events in COVID-19 patients when compared to healthy donors and influenza–ARDS patients. These findings revealed the importance of microRNAs as biomarkers that can promote more specific monitoring and treatment according to the disease pathophysiology. Moreover, the capacity of microRNAs to regulate target mRNAs may be very important in the mRNA-based vaccination field. For instance, it is well established that a single miRNA can target multiple mRNAs and that the same mRNA can be regulated by several miRNAs, which means that thousands of protein-coding genes are regulated by miRNAs [77]. So, if there are human miRNAs capable of targeting vaccine-related mRNAs or mRNAs related to the host response to these vaccines, they have the potential to be studied as biomarkers of mRNA-based vaccination response and provide more insight in this recent field.

Thus, the aim of this study is to conduct an extensive review of the known human EV-miRNAs that regulate key mRNAs involved in SARS-CoV-2 infection and vaccination response and see if they are also deregulated in the cancer context. This information will directly shed some light on how the deregulation of miRNAs caused by a cancer patient’s condition can affect their immune response capacity.

## 2. Evidence Acquisition

Using the search terms “microRNA”, “extracellular vesicles”, and “SARS-CoV-2” in PubMed, we retrieved 27 articles that were published between 2020 and 2023 for analysis. The selection of these articles was based on the relevance of their findings, particularly the interactions of miRNAs with SARS-CoV-2-related mRNAs that were experimentally validated. Among the 27 articles that were found, 20 were excluded based on exclusion criteria, as follows: (1) nonhuman miRNAs; (2) miRNAs that have no association with SARS-CoV-2-related mRNAs; (3) articles with predictive results only based on bioinformatic tools or in silico analysis of miRNA interactions with SARS-CoV-2 mRNAs; (4) protocols, comments, letters, and editorials; (5) papers included in meta-analysis or reviews.

From the seven original articles that were included in this review, the main characteristics of each article were extracted, such as the miRNA name and the validation of the interaction between the miRNA and mRNA of SARS-CoV-2, with experimental methods.

### 2.1. Literature Analysis and Evidence Synthesis

A total of 34 microRNAs were identified as being EV-miRNAs with an impact on SARS-CoV-2 infection and response. Information was extracted and synthesized from 7 original articles. Table 1 provides a summary of the included studies. Out of the articles analyzed, 3 were classified as in vitro studies, while only 2 articles focused on the analysis of patient samples. The last 2 studies incorporated both in vitro studies and the analysis of serum samples from patients, with the administration of EVs previously isolated from patients’ serum samples into cells. Most of the articles employed bioinformatic tools to identify microRNAs associated with specific conditions induced by SARS-CoV-2 infection and then validated their findings through experimental methods.

China and Germany are the primary contributors to this field with four publications in total, with each country having two publications. Additionally, there is at least one publication attributed to India, Korea, and Italy. Among these 7 publications, 2 papers focused on evaluating the effectiveness of antiviral therapeutic approaches involving EVs derived from mesenchymal cells and M2 macrophages in the treatment of COVID-19. The research in these papers aimed to investigate the potential benefits of utilizing EVs for combating the viral infection of SARS-CoV-2. On the other hand, the remaining 5 publications explored a different aspect of COVID-19 research. Specifically, these publications investigated the function of EV-derived microRNAs as biomarkers for identifying and monitoring specific complications associated with COVID-19. The objective of these studies was to understand the role of these microRNAs in the development and progression of complications related to the disease. Specifically, Mishra et al. explored the cerebral damage in COVID-19, Wang et al. correlated the low expression of a microRNA profile with elderly and diabetic patients, and three papers, Meidert et al., Borrmann et al., and Gambardella et al., investigated specific biomarkers for COVID-19 complications, such as pneumonia, ARDS, and cardiovascular disorders. Overall, these 7 publications contribute to a comprehensive understanding of COVID-19 by examining both the therapeutic potential of extracellular vesicles and the diagnostic significance of microRNAs in the context of the disease (Figure 1). However, none of the selected articles mentioned the implications of EV-derived miRNAs in SARS-CoV-2 studies in cancer prognosis, revealing a lack of investigation in this area.

Moreover, since it is through the host ACE2 receptor that SARS-CoV-2 infection begins, it was interesting to find that, in El-Shennawy et al.’s [72] research, the circulant small EVs of plasma samples of COVID-19 patients have more significant expression of ACE2 (ExoACE2) when compared to EVs of plasma samples of individuals with no SARS-CoV-2 infection. In this particular paper, it was revealed that ExoACE2, which is upregulated in the blood of COVID-19 patients, comprises part of the activation of the innate immune system, since ExoACE2 inhibits SARS-CoV-2 infection by competing with host-cell-surface ACE2, provoking an inhibition of the viral load of SARS-CoV-2 and variants, and, consequently, being part of the innate antiviral mechanism through the viral neutralization effect of human plasma.

After these findings, researchers expanded the study, investigating the differential characterization of ExoACE2 and non-ExoACE2 and their respective exo-miRNA cargo in the context of SARS-CoV-2 infection. In these studies, Mimmi et al. [85] reported that ExoACE2 has in its cargo an upregulation and downregulation of specific microRNAs, mentioned in Table 1, considered as potential biomarkers that can improve and clarify the immune response of patients against COVID-19. Moreover, Latini et al. [86] also reported that epigenetic mechanisms of miRNAs are involved in the expression of ACE2. Particularly, the downregulation of an miRNA that belongs to a member of the let-7 family in COVID-19 patients promotes an overexpression of ACE2. The downregulation of this miRNA is present in other comorbidities such as diabetes mellitus 2 and inflammation [87], making the correlation of these conditions a promising field of investigation.

### 2.2. SARS-CoV-2-Related EV-miRNAs and Their Potential Impact in Cancer Patients’ Immune Response to Vaccination

Interestingly, we observed that two miRNAs, hsa-miR-24-3p and hsa-miR-145-5p, were mentioned in more than one of the selected articles in this review (Table 1) [79,82,83]. Moreover, both miRNAs were also capable of targeting the 3′-UTR of SARS-CoV-2 S mRNA, which made them more promising for further study as potential biomarkers of response to mRNA vaccines in the future.

These two miRNAs were previously identified as regulators of multiple targets present in cancer progression and, separately, in SARS-CoV-2 infection [88,89,90,91,92,93]. Therefore, miR-24-3p and miR-145-5p can have the potential to achieve a synergic effect on cell phenotypes in cancer patients with SARS-CoV-2 infection, promoting a more aggressive state of both diseases [94]. Moreover, the deregulation of two miRNAs that bind to the mRNA of the SARS-CoV-2 spike protein can also have an impact on the patient’s response to vaccination, more specifically the mRNA vaccines. In fact, hsa-miR-24-3p and hsa-miR-145-5p are widely studied in cancer development [95,96]. The overexpression of miR-24-3p is identified in the risk of poor prognosis in various human carcinomas, such as hepatocellular carcinoma and lung cancer, with an important role in cellular proliferation, metastasis promotion, and inhibition of apoptosis [97,98]. In contrast, the expression of miR-145-5p is downregulated in a wide range of cancers, including colorectal cancer, non-small-cell lung cancer, breast cancer, and prostate cancer, as this miRNA is mainly a tumor suppressor, regulating epithelial-to-mesenchymal transition (EMT) and inhibiting cancer stem cells; tumoral growth, invasion, and metastasis; and tumor angiogenesis [99,100,101]. However, it is important to study the expression of these EV-miRNAs in cancer patients with COVID-19 disease simultaneously and before and after they receive vaccination, in order to understand how their expression impacts the patients’ immune response.

Since hsa-miR-24-3p and hsa-miR-145-5p are dysregulated both in cancer and in SARS-CoV-2 infection disease, the synergic effects of these events can promote the exacerbation of negative health outcomes in cancer patients’ response to viral infection and vaccination. Therefore, it is important to investigate the correlation of the levels of these EV-miRNAs with cancer patients’ response to mRNA vaccines to define a molecular profile that will allow us to stratify patients into good or bad responders and take the necessary precautions to improve their treatments and follow-ups (Figure 2).

## 3. Discussion

Previously, EVs were considered to be cell waste with no molecular interest, a scenario that has completely changed with recent advances in knowledge about these nanoparticles. In the last decade, EVs have been recognized as an intrinsic and crucial mechanism of intercellular communication, allowing cells to exchange proteins, lipids, and genetic material, such as microRNAs, and they are one of the current main focuses in scientific research. However, little is known about the influence of EVs in SARS-CoV-2 infection. The current information published in this area, mainly focused on in vitro investigations, alongside research involving human liquid biopsies, evidences the clear importance of these molecular transporters and their cargo in the pathophysiology of this viral infection. Nevertheless, studies are still needed to further explore the implications of the microRNAs of these EVs in COVID-19, specifically in cancer patients.

Thus, future investigations of EV-derived microRNAs in larger cohorts are essential to deepen and add more knowledge in this field, since this review establishes the relevance and applicability of EV-derived microRNAs in the SARS-CoV-2 response to this viral infection as delivery agents in immune response and important transporters in the process of vaccination against COVID-19, particularly in mRNA vaccines. In addition to that, this review reveals that there are no studies on EV-derived microRNAs in SARS-CoV-2 in cancer investigation, exposing an urgent requirement of additional studies in this area. Therefore, in the domain of research on the subsequent phases of EV-derived microRNAs in cancer patients who were subject to previous or still have an ongoing SARS-CoV-2 infection and/or COVID-19 vaccination, the study of the two EV-miRNAs identified in this review, hsa-miR-24-3p and hsa-miR-145-5p, should be conducted, since these two miRNAs were found to play a vital role in the molecular pathways of both mentioned diseases. From this perspective, we will be able to understand whether the dysregulation of these microRNAs already identified in cancer progression will trigger a different immune response against SARS-CoV-2 infection or vaccination in cancer patients.

## 4. Conclusions

This review highlights the investigation of the role of EV-miRNAs in COVID-19, with a specific emphasis on the domain of knowledge in cancer research, to identify the current information that exists in this field and the main research stages to explore in further investigation. This review confirms the importance of studying the expression of these microRNAs in cancer patients who have been diagnosed with SARS-CoV-2 infection or vaccinated with COVID-19 vaccines, with the purpose of understanding the molecular implications that the upregulation or downregulation of these microRNAs provoke in signalling pathways that are involved in cancer and in SARS-CoV-2 simultaneously. Moreover, further studies are needed to comprehend the use of EV-derived miRNAs from plasma or other body fluids in clinical practice.

Moving forward, future research in this field should focus on investigating the role of two EV-derived miRNAs, namely, hsa-miR-24-3p and hsa-miR-145-5p, in the molecular pathways underlying the synergistic effects that promote a more aggressive state in cancer, SARS-CoV-2 infection, and/or COVID-19 vaccination.

In conclusion, the identification of functional EV-derived miRNAs in the context of SARS-CoV-2 infection has significant implications for the use of these miRNAs as predictive biomarkers for assessing the immune response in cancer patients. These miRNAs can serve as biomolecular indicators of how the immune system responds to both viral infection and cancer progression. Understanding the molecular networks associated with these EV-derived miRNAs is crucial to discover their specific roles and mechanisms of action. This knowledge has the potential to provide valuable factors for predicting the severity of viral infections and the progression of cancer in affected individuals. Importantly, the integration of this information into clinical practice can lead to more personalized treatment approaches for cancer patients. By considering the status of these EV-derived miRNAs, healthcare professionals can consider treatment strategies for individual patients based on their specific immune response profiles. This personalized approach holds the promise of improving treatment outcomes and overall patient care in the context of both viral infections and cancer.

## Figures and Tables

**Figure 1 cancers-15-04017-f001:**
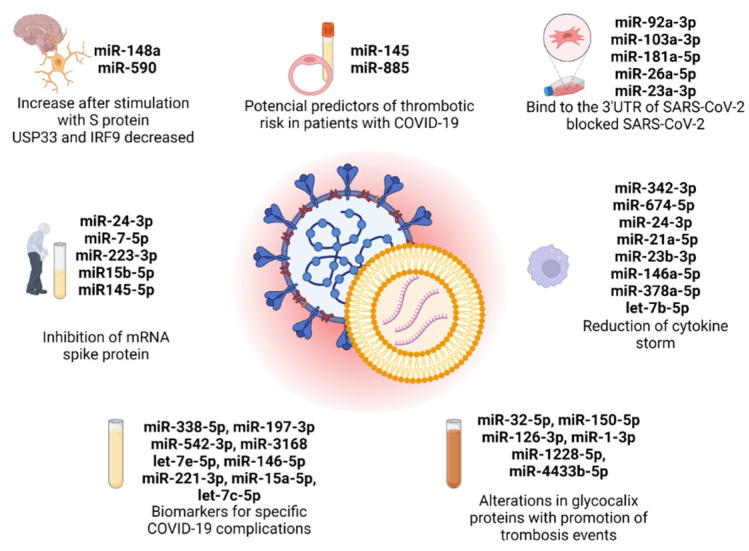
EV-derived microRNAs present in the 7 original articles selected in this review, with the principal conclusion in each study. Created with BioRender.com (accessed on 10 July 2023).

**Figure 2 cancers-15-04017-f002:**
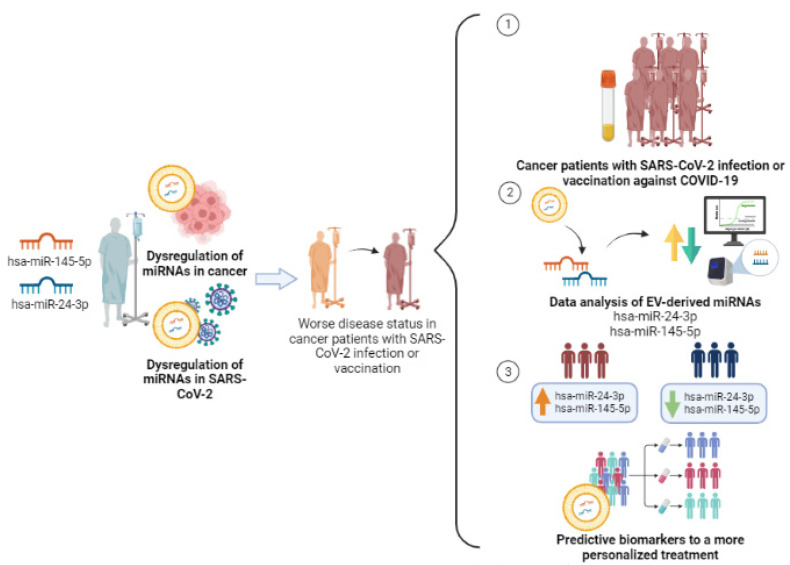
Study design of the two selected miRNAs, hsa-miR-24-3p and hsa-miR-145-5p, in cancer patients with SARS-CoV-2 infection and/or COVID-19 vaccination for predictive biomarkers in a more personalized treatment. Created with BioRender.com (accessed on 10 July 2023).

**Table 1 cancers-15-04017-t001:** The seven publications selected in this review, with the microRNAs explored, the type of samples, and the principal results of each study, respectively. * Additional articles with interest in investigation but not found in PubMed with search terms.

miRNA	Samples/EV Sources	Results	Reference
miR-148a and miR-590	HEK-293T and human microglial cell line (CHME3)	Neuroinflammation induced by miR-148a and miR-590 derived by EVs after S protein stimulation. Hyperactivation of microglia, with a reduction in USP33 and IRF9 expression.	Mishra and Banerjea [78]
miR-7-5p, miR-24-3p, miR-223-3p, miR-145-5p, and miR-15b-5p	Two cohorts: 30 serum samples per group (4 groups) in cohort 1 and 20 serum samples per group (4 groups); HEK293T cells	miRNAs that are low in elderly and diabetic patients inhibited S protein replication. The expression of these miRNAs increased after long periods of exercise.	Wang et al. [79]
miR-23a-3p, miR-26a-5p, miR-92a-3p, miR-103a-3p, and miR-181a-5p	Neuronal stem cells; human lung fibroblasts cell line LL24; human bronchial epithelial cell line Beas-2B; mouse microglial cell line BV2; human neuroblastoma cell line SK-N-BE(2)C; Vero cells	miRNAs of MSC-EVs inhibit viral replication in 3′UTR of SARS-CoV-2 genome (conserved region).	Park et al. [80]
miR-338-5p, miR-197-3p, miR-542-3p, miR-3168, let-7e-5p, miR-146a-5p, miR-221-3p, miR-15a-5p, and let-7c-5p	100 blood samples: 20 patients with COVID-19 pneumonia, 20 patients with COVID-19 ARDS, 20 healthy donors, 28 patients with sepsis associated with ARDS, and 12 patients with bacterial community-acquired pneumonia	miR-338-5p targeted IL6 and OR52N2. Upregulated miR-542-3p increased furin activity. Downregulation of miR-3168 and let-7e-5p increased CXCL8 levels in severe COVID-19 infection. Inhibition of TLR4 by miR-146-5p in pneumonia and by let-7e-5p in COVID-19 ARDS. Downregulation of cytokines by miR-221-3p and miR-15a-5p in immunosuppressive state of COVID-19.	Meidert et al. [81]
miR-342-3p, miR-674-5p, miR-24-3p, miR-21a-5p, miR-23b-3p, miR-146-5p, miR-378a-5p, and let-7b-5p	Primary mouse peritoneal macrophages M2 (mMφ) isolated from peritoneal dialysis (PD)	Peritoneal M2-EVs promote reduction of pro-inflammatory cytokines levels.	Wang et al. [82]
miR-145 and miR-885	26 serum samples and umbilical vein endothelial cells (HUVECs)	HUVECs treated with serum of COVID-19 patients induce a reduction in miR-145 and miR-885 and an overexpression of tissue factor and Willebrand factor in endothelial cells.	Gambardella et al. [83]
miR-32-5p, miR-150-5p, miR-126-3p, miR-1-3p, miR-1228-5p, and miR-4433b-5p	60 blood samples of COVID-19 patients	Increase in glycocalyx components and decrease in ADAMTS13 by downregulation of microRNAs.	Borrmann et al. [84]
let-7g-5p, miR-4454+miR-7975, hsa-miR-208a-3p, and hsa-miR-323-3p	Plasma samples of COVID-19 patients and cell culture (HEK-293 and HeLa cells) [72]; serum samples of 6 COVID-19 patients [85]	ExoACE2 with upregulation of let-7g-5p and hsa-miR-4454+miR-7975 and downregulation of hsa-miR-208a-3p and hsa-miR-323-3p, compared to non-ACE2-expressing exosomes.	El-Shennawy et al. [72], Mimmi et al. [85] *
hsa-let7b- 5p	HeLa cell line; 60 nasopharyngeal swabs (NPS) of COVID-19 patients	Low expression of hsa-let7b-5p in COVID-19 patients leads to a lack of regulation of genes (ACE2 and DPP4) exploited by SARS-CoV-2.	Latini et al. [86] *

## Data Availability

The data can be shared up on request.

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
