# Peer review of "MicroRNAs Derived from Extracellular Vesicles: Keys to Understanding SARS-CoV-2 Vaccination Response in Cancer Patients?"

_cancers, 2023, doi:10.3390/cancers15164017_

Round 1
Reviewer 1 Report (Previous Reviewer 1)
In this revised version, authors have added few references supporting their claims. however, once again, it is not clear whether it is a research or a review article. Materials and methods section has not clearly described comprehensively. They have discussed others' work in this section which is inappropriate. They should rather remove this part and present it in the form of comprehensive review on role of EV-derived miRNAs in COVID-vaccinated cancer patients.
Moderate editing of English language (grammar) is required
Author Response
We would like to thank the reviewer for the feedback and suggestions for the improvement of our manuscript.
All the alterations in the manuscript were made using Microsoft Word "track changes" feature and are displayed in the manuscript.
According to the reviewer suggestions, we removed the section of the bioinformatic analysis of the 2 microRNAs that we found to be the most interesting to look at in the cancer context plus SARS-CoV-2 infection. We hope that these alterations make the review article more clear to the reviewer.
We also revised and improved the english and grammar of the manuscript.
Reviewer 2 Report (Previous Reviewer 2)
The reviewer thanks the authors for their responses back to the reviewer’s comments and the corresponding revisions to the manuscript.
Minor editing of English language required
Author Response
We would like to thank the reviewer for the feedback and suggestions.
We revised the paper once more to check and correct typos and english editing.
All the alterations in the manuscript were made using Word "track changes" feature and are displayed in the manuscript.

Round 2
Reviewer 1 Report (Previous Reviewer 1)
The manuscript still has Materials/methods and results as separate section which is not required. It should be presented as continuous text with subject-specific subheadings.
Author Response
According to the reviewer suggestions, we merged the materials and methods and results section into one section, entitled "2. Evidence Acquisition".
All the alterations are highlighted in the manuscript using Mircrosoft Word "track changes" feature.
This manuscript is a resubmission of an earlier submission. The following is a list of the peer review reports and author responses from that submission.
Round 1
Reviewer 1 Report
Although this manuscript is claimed to be a review article, it presents some bioinformatic analysis of miRNAs implicated in COVID-19. In the first section it explains the biology of coronavirus and pathophysiology of COVID-19 and strategies for vaccine development quite well. Further, it also presents an overview of EV-miRNA as potential biomarkers to COVID-19 vaccines in cancer patients. In the next sections, it presents results based on literature search (7 articles). miRNAs implicated in COVID-19 in are listed in a table. Same miRNAs and their mechanism of action have been explained further which is redundant. Further, the study was focused only on 2 miRNAs which were common in two articles (not cited with proper references) for which they have identified protein targets using in silico analysis. They have verified these targets using experimental methods – the results for which have not been shown in this article (these should have been shown). The authors have also claimed that these same miRNAs are also implicated in cancer patients with COVID-19 infection but once again without citing correct references. Further, the authors have constructed a protein-protein interaction network for these targets and carried out their gene ontology analysis which is not very conclusive.
Moderate english editing and section heading revision is required.
Reviewer 2 Report
The authors present an interesting review summarizing the state of the art concerning the role of EVs-derived miRNAs in COVID-19 infection and the potential use of EV-miRNAs as prognosis biomarkers for vaccination response in cancer patients. Although the intent is really fascinating, the authors missed some recently reported pieces of shreds of evidence focused on the role of ACE2 expressing EV. Indeed in 2022, El-Shennawy et al. reported an increase in circulating exosomes expressing ACE2 (ExoACE2) in the plasma of COVID-19 patients, highlighting the potential role of a first-line immune response against viral infection [Shennawy L, Hoffmann AD, Dashzeveg NK,Mc Andrews KM, Mehl PJ, Cornish D, et al. Circulating ACE2-expressing extracellular vesicles block broad strains of SARS-CoV-2. Nat Commun 2022;13:405.]. Taking advantage of these findings, the molecular characterization of both ExoACE2 and non-ExoACE2 was recently performed, and a differential miRNA signature between the two exosomes subpopulation was reported [Mimmi S et al. SARS CoV-2 spike protein-guided exosome isolation facilitates detection of potential miRNA biomarkers in COVID-19 infections. Clin Chem Lab Med. 2023]. Other COVID19 associated miRNAs were also reported:
- Latini, A, Vancheri, C, Amati, F, Morini, E, Grelli, S, Claudia, M, et al.. Expression analysis of miRNA hsa-let7b-5p in naso-oropharyngeal swabs of COVID-19 patients supports its role in regulating ACE2 and DPP4 receptors. J Cell Mol Med 2022;26:4940–8.
- Garg, A, Seeliger, B, Derda, AA, Xiao, K, Gietz, A, Scherf, K, et al.. Circulating cardiovascular microRNAs in critically ill COVID-19 patients. Eur J Heart Failure 2021;23:468–75.
- Fernández-Pato, A, Virseda-Berdices, A, Resino, S, Ryan, P, Martínez-González, O, Pérez-García, F, et al.. Plasma miRNA profile at COVID-19 onset predicts severity status and mortality. Emerging Microbes Infect 2022;11:676–88.
The authors should take into consideration these relevant advances in the definition of a prognostic COVID-19-related miRNA profile.
Good luck.
some stylistic and typing improvement needed